# Arterial Infusion and Isolated Perfusion in Combination with Reversible Electroporation for Locally Relapsed Unresectable Breast Cancer

**DOI:** 10.3390/cancers16233991

**Published:** 2024-11-28

**Authors:** Kornelia Aigner, Emir Selak, Monika Pizon, Karl Reinhard Aigner

**Affiliations:** 1Department Tumor Biology, Medias Klinikum, 84489 Burghausen, Germany; 2Department Surgical Oncology, Medias Klinikum, 84489 Burghausen, Germany; e.selak@medias-klinikum.de (E.S.); info@prof-aigner.de (K.R.A.); 3Transfusionsmedizinisches Zentrum, 95448 Bayreuth, Germany; mpizon@simfo.de

**Keywords:** triple-negative breast cancer, unresectable tumor mass, electrochemotherapy, arterial infusion, isolated perfusion chemotherapy

## Abstract

Although in most cases breast cancer is curable, some specific subtypes and late stages of the disease are difficult to treat, and treatment-related side effects often are severe. Two methods that focus on the local treatment of breast cancer, regional chemotherapy and electroporation, can be combined to increase the treatment efficacy at the tumor site and decrease systemic side effects. This publication explains the technique, systemic impact on circulating tumor cells, and first results of this treatment approach.

## 1. Introduction

Breast cancer treatment is well established and many different treatments are offered for different breast cancer subtypes. The prognosis is relatively good for hormone receptor-positive breast cancer with a 10-year recurrence-free survival rate of 85–90% for cancers diagnosed early. Treatment options and prognosis for HER2 (human epidermal growth factor receptor 2)-positive breast cancer has improved over the last decade with the availability of advanced HER2-targeted treatments and antibody–drug conjugates.

However, for triple-negative breast cancer (TNBC), also referred to as basal breast cancer, treatment options are limited. About 15% of breast cancer cases are triple negative. TNBC growth is not dependent on hormone receptor stimulation and therefore cannot be stopped by hormone blockade. Furthermore, TNBC has a normal or decreased expression of HER2 and therefore is also not responsive to HER2-targeted treatment. Other cell surface molecules are currently under investigation for their eligibilty as treatment targets. TROP2 is a potential target for antibody–drug conjugates in TNBC treatment [1,2,3]. Other surface molecules, like EGFR, might be potential targets [4,5], and for some TNBCs with a specific tumor microenvironment, immunotherapy may play a role as well [6,7,8,9]. However, TNBC seems to be a heterogenous subgroup of breast cancers, and the only attribute they have in common is that their growth is not dependent on hormone receptors nor HER2. In particular, the tumor microenvironment and immune cell invasion into the tumor tissue is crucial to the treatment response to chemotherapeutics and/or immunotherapies like checkpoint inhibition. Advanced TNBCs in particular have unsatisfying treatment responses, indicating a need for new treatment options. The latest studies of recurrent or metastasized TNBC show overall survival rates of 8–13 months. Some patients have oligo- or multiple metastatic disease and have to be treated systemically. If these patients suffer from exhaustion and immune suppression due to multiple treatment lines, focal locoregional treatment of painful and/or life-threatening metastases can be considered. For liver metastases from breast cancer, trans-arterial chemoembolization is recommended by guidelines and offers an effective method that maintains quality of life for these patients.

Although not as life-threatening as liver metastases, fast-growing recurrences at the primary tumor site can be painful and cause psychological distress, especially when the tumor size surpasses 5 cm or involves ulceration and bleeding. These chemoresistant, progressive, or recurrent tumors at the primary site frequently do no longer respond to drug concentrations that can be reached maximally by systemic chemotherapy. Some patients do not benefit sufficiently from multiple systemic treatment options and relapse, develop multiresistant disease, are exhausted after multiple treatment lines, or refuse further systemic treatments. Therefore, a focused, locally highly concentrated treatment is amenable for both metastatic diseases, including primary recurrences and primary recurrences without metastatic burden but with chemoresistant, unresectable, ulcerated, bleeding, or otherwise distressing tumors.

Reversible electroporation has been used for breast cancer in several studies with promising results [10,11,12,13,14,15,16]. The electric voltage that is applied to the tumor lesion temporarily causes cell membrane poration and therefore increases drug uptake into the tumor cells multifold (Figure 1A). Usually systemic chemotherapy with bleomycin is combined with the therapy. Regional chemotherapy, either arterial infusion [17,18,19,20,21] or isolated perfusion [22,23,24], is used to maximize local drug concentration and efficacy and minimize systemic side effects. The combination of both therapies intends to increase local drug uptake into the tumor cells. The aim of this work is a proof of concept for the combination of reversible electroporation with arterial infusion and isolated perfusion. The two techniques have not been combined before. Technical feasibility, safety, and local response are evaluated. In particular, the constellation of drug exposure times, dosages, and electroporation durations were examined to analyze the potential danger of tumor cell release during exposure to electric voltage. Cell adhesion could be decreased due to electric forces and potentially cause an increased risk of metastasis if combined with only low drug dosages in the arterial, and hence localized, setting. This potential risk was observed via circulating tumor cell surveillance.

## 2. Materials and Methods

**Reversible electroporation:** An IGEA cliniporator with a hexagonal fixed needle constellation (H-30-ST, or H-40-ST) was used for cutaneous electroporation. Depending on the ventrodorsal tumor size, needle lengths of 30 or 40 mm were used. During the short window of high drug exposure in the tumor region, the tumor region at the breast was treated with 4–23 implementations of needle setting and electric pulses depending on the size of the lesion. The application fields of the hexagonal electrodes overlapped to ensure complete treatment of the tumor area (Figure 1B).

**Arterial infusion chemotherapy:** For the treatment of locally relapsed breast cancer, arterial short-term infusion into the mammarian/subclavian artery was performed. A triple combination chemotherapy of cisplatin, doxorubicin, and mitomycin with low total dosages was infused over 5–7 min. The time of high drug exposure during short-term arterial infusion was 5–7 min. Total drug dosages were 30–40 mg cisplatin, 20 mg doxorubicin, 10 mg mitomycin.

**Isolated thoracic perfusion chemotherapy:** Isolated thoracic perfusion extends the drug exposure time and enlarges the region of high drug exposure compared to mammarian/subclavian artery infusion only. The thoracic region was isolated with occlusion catheters in the vena cava and in the aorta with the balloons placed right beneath the diaphragm [22,23,24]. Pneumatic cuffs at the upper arms further reduced the treatment volume during the procedure. The natural blood flow enables circulation inside the isolated thoracic region (Figure 2). Chemotherapeutic agents were either infused via the infusion line of the occlusion catheter that ended at the tip of the catheter above the balloon or via an additional angiographic catheter, as described in the section for arterial infusion above. Isolated perfusion was performed for 15 min before the balloons were deflated and the systemic blood flow was restored. The amount of time of high drug exposure during the isolated thoracic perfusion was 15 min. Total drug dosages were 60–90 mg cisplatin, 30–40 mg doxorubicin, 10–20 mg mitomycin.

**Drugs:** A three-drug combination of cisplatin, doxorubicin, and mitomycin was used for arterial infusion only and in combination with isolated perfusion [22,23,24].

**Patients:** Patients with either relapsed and/or chemoresistant breast cancer and an unresectable tumor mass at the primary location were treated. The tumors caused either pain, bleeding, or psychological distress and therefore were the reason for increased treatment pressure. Distant metastases were not exclusion criteria as long as they were not acutely life threatening.

**Treatment modality:** The procedure was undertaken under full anesthesia. Either arterial infusion or arterial infusion with isolated thoracic perfusion was performed. Each procedure was combined with reversible electroporation of the tumor region. Each procedure was followed by chemofiltration to reduce systemic drug exposure to a minimum for the heavily pretreated and exhausted patients [22].

**Circulating Tumor Cells:** The term CETC/CTC was chosen to emphasize that the maintrac analysis method specifically detects and quantifies circulating **epithelial** tumor cells in the blood. In contrast, the abbreviation CTC can also refer to tumor cells from other origins, including hematological tumors, but frequently is used in the literature.

Before and 24 h after the procedure, 15 mL of EDTA blood was drawn and analyzed to quantify circulating tumor cells (CETCs/CTCs) using the maintrac approach [25]. Briefly, after red blood cell lysis, an immunofluorescence assay with a FITC (fluorescin isothiocyanate)-conjugated EpCAM (epithelial cell adhesion molecule) antibody was employed to identify CETCs/CTCs. An isotypic control was performed to determine background staining levels, and propidium iodide (PI) staining was used to differentiate living and dead cells. Red and green fluorescence analysis was conducted using a Fluorescence Scanning Microscope (ScanR, Olympus, Germany) for the visual examination of EpCAM-positive cells. Vital CETCs/CTCs were defined as EpCAM-positive cells with intact morphology and no nuclear PI staining. Only these cells were counted for analysis (Figure 3) [22,23,24,25,26,27,28,29].

## 3. Results

### 3.1. Patients and Treatments

Patients with chemoresistant or locally recurrent FIGO (International Federation of Gynecology and Obstetrics) stage III or IV breast cancers were treated. The median age was 51 years; 12 patients suffered from triple-negative disease, and 2 were resistant to anti-hormonal therapy. The median size of the treated lesions was 7.6 cm (SD3.3 cm), and five patients had ulcerated lesions. Eight patients had distant metastases (liver, bones, lung) that were treated with regional chemotherapy in further treatment cycles without electroporation. A total of 21 treatment cycles were performed for 14 patients, and all included arterial infusion and reversible electroporation. Six cycles were performed with additional isolated thoracic perfusion. All procedures concluded without technical or medical problems (Figure 4A).

### 3.2. Distant Disease

Distant disease was present in 8 out of 14 patients. Since systemic treatment was either not possible due to patient exhaustion (n = 2), multiple chemo- and immuneresistance (n = 4), or refusal by the patient (n = 2), these patients were additionally treated with isolated perfusions to other body regions (hypoxic isolated abdominal perfusion, hypoxic isolated pelvic perfusion) or TACE (trans-arterial chemoembolization).

### 3.3. Circulating Tumor Cells (CETCs/CTCs)

In 14 out of 21 cycles, CETCs/CTCs were detected directly before the treatment started. We compared the number of CETCs/CTCs before and 24 h after the treatment for each cycle individually. In 11 of the 21 cycles, CETCs/CTCs decreased 24 h after the therapy, in one case CETCs/CTCs stayed at a low and stable level, and in two cases cell numbers increased after 24 h. The two cases with an increase in cell numbers were the same in that electroporation duration exceeded the time of high drug exposure. One of these patients, however, had a local tumor shrinkage that led to resectability with free margins, but deteriorated 2 months later with massive liver metastases. In seven cases where no CETCs/CTCs were detected before the treatment, they remained negative for CETCs/CTCs 24 h after the procedure (Figure 5).

### 3.4. Local Response Rates

Tumor shrinkage was observed in all but one case 2–7 days after each of the 21 cycles. Radiologic controls 4–8 weeks after the treatment showed a median of 45% shrinkage in the largest tumor’s diameter and showed complete response in two cases and R0 resectability in four cases (Figure 4B).

### 3.5. Overall Survival

Median overall survival for all patients was 13 months. Stage IV patients had a median overall survival period of 10.5 months and stage III patients had a median overall survival period of 17.5 months (Figure 4B).

### 3.6. Adverse Effects

Typical adverse events to chemotherapy were not observed, or were observed only to a very mild extent. Neutropenia never exceeded grade 2 and no patient suffered from hand-foot syndrome during treatment or the follow-up period. Mild nausea occurred rarely, but disappeared within two days after treatment. Hair loss only occurred in one case. Wound infection at the catheter insertion site occurred in one case. Skin discoloration due to electrochemotherapy was not observed, which might be due to the use of cisplatin instead of bleomycin. Other side effects did not occur either.

## 4. Discussion

Two methods, reversible electroporation and arterial infusion chemotherapy, both intending to maximize local effects and minimize side effects, were combined successfully in this pilot study. Breast cancer in general can be treated by multiple different treatment modalities, with a great variety of chemotherapeutic, immune modulating, and molecular targeting drugs for each specific kind of subgroup (HER2-positive, HER2-low, hormone receptor-positive, and triple-negative breast cancer). However, quite a number of patients still experience multiple drug resistance and/or severe side effects that shift the balance of benefits versus side effects of systemic treatments to the unfavorable side. If those patients suffer from large tumor lesions that either cause pain or psychological distress, a local treatment with a potential for quick response and local tumor shrinkage should be considered.

Electrochemotherapy is known to enhance the local efficacy of systemically applicated drugs. In particular, the efficacy of bleomycin is multiplied 80-fold, while that of cisplatin is multiplied 8-fold [30,31,32]. Bleomycin, therefore, is the most-used drug during reversible electroporation. The systemic efficacy of bleomycin is neglectable in terms of tumor response to distant side as well as in terms of side effects because of the reduced dosage and very low concentration during electrochemotherapy.

Arterial infusion is mostly known for its use in liver treatments, but also is used for local treatment of breast cancer. Isolated perfusion chemotherapy is mostly known for its use in isolated liver or limb perfusion, but also has been conducted for advanced breast cancer and other different entities located in the thoracic region (isolated thoracic perfusion) [22,23,24], abdominal region (isolated hypoxic abdominal perfusion) [33], or pelvic region (isolated hypoxic pelvic perfusion) [34].

Since the intention of a treatment with electrochemotherapy is not to generate a systemic effect, and the technique for regional chemotherapy is feasible, it is a natural conclusion to combine reversible electroporation with arterial infusion instead of intravenous drug delivery.

Arterial infusion allows for a multifold drug concentration at the tumor due to the first-pass effect. It is clearly of benefit to use this short time window of high concentration during the electroporation process. However, for large tumor masses, where multiple needle placements are needed and the procedure takes longer than the short-term infusion, isolated thoracic perfusion (ITP) should be considered. ITP also delivers better safety when needles cannot be placed optimally and the lesion cannot be completely covered by electric voltage. For those cases, high drug exposure of the lesion as well as the surrounding area might avoid residual tumor remnants.

Both arterial infusion and isolated perfusion chemotherapy enhance local drug concentration and have a greater potential for local efficacy. If, for isolated perfusion, the 20–40-fold higher cisplatin concentrations equal or outperform the efficacy increase of bleomycin, electrochemotherapy needs to be investigated.

Reversible electroporation is mostly combined with systemic bleomycin, but it also increases cisplatin efficacy compared to its use without cell membrane poration. For arterial infusion and isolated abdominal, pelvic, or limb perfusion chemotherapy, drugs that also work under hypoxic conditions like cisplatin, doxorubicin, and mitomycin are mostly used worldwide. Although during isolated thoracic perfusion the physiologic milieu does not become hypoxic, the established knowledge of dosing for reduced treatment volumes led to the choice of using the same drugs for isolated thoracic perfusions.

The aim of this pilot study was to investigate the feasibility and safety of the combination of reversible electroporation and regional chemotherapy for advanced breast cancer patients with unresectable local recurrent tumor lesions. All procedures concluded smoothly from a technical standpoint and did not result in medical problems, similar to the outcomes for arterial infusion and ITP conducted separately in the same department. The concern for a possible induced metastatic risk was not confirmed, as the number of circulating tumor cells did not increase in 19 out of 21 cases. In the two cases where electroporation lasted longer than high drug exposure was ensured by arterial infusion, the CETCs/CTCs increased 24 h after the procedure. This shows the potential risk and the need for very good planning of the procedure, especially when the total dosage of the drug is lower than for systemic chemotherapy alone.

For cases where the tumor mass is bigger than 5 cm or difficult to reach with the electrodes, electroporation zones cannot cover the entire lesion sufficiently, or satellite metastases are present, its combination with isolated perfusion is preferable.

The limitations of this study are its small number of patients and the heterogeneity of the patients in terms of tumor size and metastatic lesions. However, the results of this pilot study encourage plans for clinical trials.

## 5. Conclusions

The combination of reversible electroporation with intra-arterial chemotherapy or isolated perfusion seems to be feasible and to result in a good clinical response with neglectable side effects. The treatment seems to be repeatable and can lead to resectability. However, for specific cases, the use of isolated perfusion might be more effective and safer in terms of tumor coverage and avoiding increased metastasis risk.

## Figures and Tables

**Figure 1 cancers-16-03991-f001:**
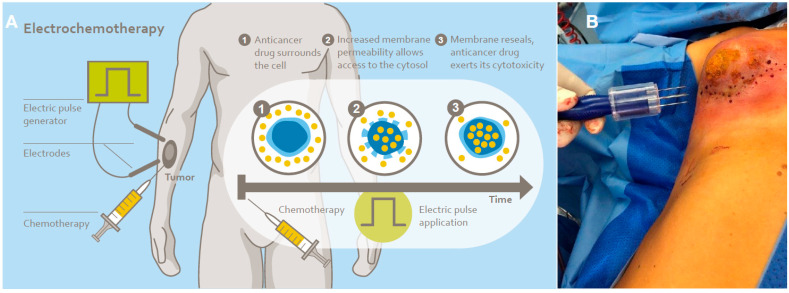
(**A**): Mechanism of reversible electroporation with chemotherapy, also known as electrochemotherapy (**B**): Reversible electroporation of a locally recurrent unresectable breast cancer with the hexagonal 30 mm electrode and the cliniporator. The procedure was combined with isolated thoracic perfusion to cover the complete lesion with infiltration of the chest wall.

**Figure 2 cancers-16-03991-f002:**
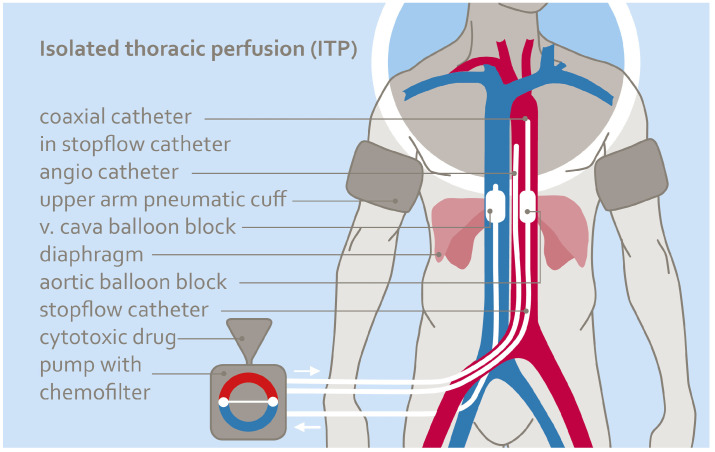
Isolated thoracic perfusion is a technique that isolates and temporarily decouples the thoracic region from the rest of the body’s blood flow. Isolation is performed using balloon perfusion catheters in the vena cava and aorta and pneumatic cuffs at the upper arms. The isolated area is connected to an external circuit consisting of a roller pump and hemofiltration machine.

**Figure 3 cancers-16-03991-f003:**
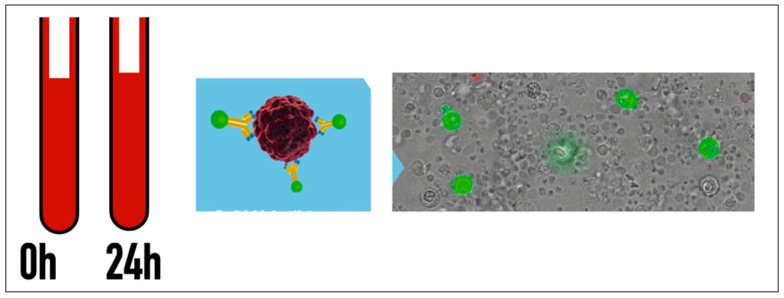
Circulating Tumor Cell (CETC/CTC) detection: Blood was analyzed before and 24 h after each procedure. CETC/sCTCs were selected depending on positivity of the surface marker EPCAM (epithelial cell adhesion molecule) and negativity for CD45 (CD: cluster of differentiation). CETCs/CTCs were labeled with a fluorescent EPCAM antibody and counted under microscopy.

**Figure 4 cancers-16-03991-f004:**
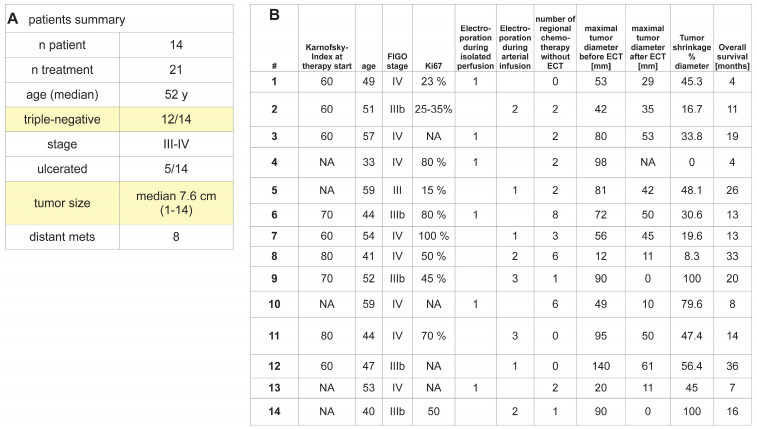
**Patients and treatment:** A total of 14 patients with the disease characteristics described in (**A**) were treated in 21 treatment cycles. Individual patient characteristics and treatment responses are described in (**B**). Treatment consisted either of arterial infusion into the mammarian/subclavian artery or the aforementioned treatment with the addition of isolated thoracic perfusion. All treatments were combined with reversible electroporation.

**Figure 5 cancers-16-03991-f005:**
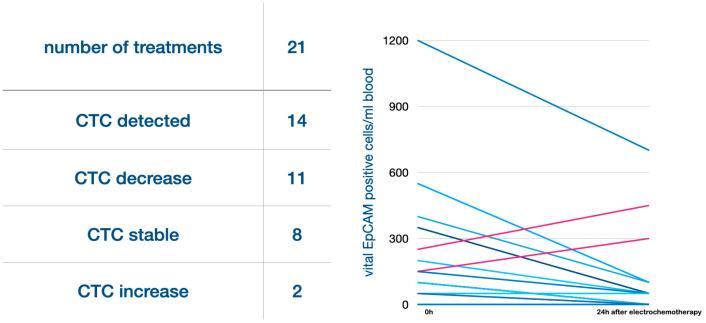
Results for CETCs/CTCs: Of 21 treatments, in 14 cases CETCs/CTCs were detected. Of these, in 11 cases the number of CETCs/CTCs decreased 24 h after the treatment, 2 had an increase, and 1 was stable. In the seven cases where no CETCs/CTCs were detected before the treatment, CETCs/CTCs were also not detected 24 h after the treatment. In total, eight cases had stable CTC numbers.

## Data Availability

The raw data supporting the results are available via email. Please contact the corresponding author.

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
