# Peer review of "Arterial Infusion and Isolated Perfusion in Combination with Reversible Electroporation for Locally Relapsed Unresectable Breast Cancer"

_cancers, 2024, doi:10.3390/cancers16233991_

Round 1
Reviewer 1 Report
Comments and Suggestions for Authors
1. What is the rate of prognosis for breast cancer cases among German populations?
2. Which method you are using for quantifying circulating tumor cells.
3. How you have selected these EPCAM positive cells.
Author Response
-
What is the rate of prognosis for breast cancer cases among German populations? Answer: Since breast cancer occurs in several subtypes with very divergent prognosis, we only answered this question for the advanced, recurrent, or metastasized triple negative breast cancer. Additionally we dscribed the difference between the subtypes and the reason for the bad prognosis. These changes were done in the introduction.
- Which method you are using for quantifying circulating tumor cells? Answer: The used method for quantifying circulating tumor cells is the maintrac method. We included a more detailed description of the method in the material and method section.
- How you have selected these EPCAM positive cells. Answer: EpCam positive cells have been selected as described in the added detailed description, which was added to the material and methods section: „Before and 24 hours after the procedure, 15 ml of EDTA blood was drawn and analyzed to quantify circulating tumor cells (CETC/CTC) using the maintrac® approach [7]. Briefly, after red blood cell lysis, an immunofluorescence assay with a FITC (fluorescin isothiocyanate)-conjugated EpCAM (epithelial cell adhesion molecule) antibody was employed to identify CETC/CTC. An isotypic control was performed to determine background staining levels, and propidium iodide (PI) staining was used to differentiate living and dead cells. Red and green fluorescence analysis was conducted using a Fluorescence Scanning Microscope (ScanR, Olympus, Germany) for the visual examination of EpCAM-positive cells. Vital CETC/CTC were defined as EpCAM-positive cells with intact morphology and no nuclear PI staining. Only these cells were counted for analysis“
Reviewer 2 Report
Comments and Suggestions for Authors
The manuscript presented for review is meant to be a "proof-of-concept" article describing the effect of combined reversible electroporation and chemotherapy administered by arterial infusion or isolated perfusion in locally relapsed unresectable breast cancer.
The method presented is intriguing and may provide a safe solution for selected patients in cases where other therapeutic options have been depleted.
My suggestions for improvement are as follows:
1. Minor English proofing needed
2. Please decide on just one abbreviation for circulating tumor cells and use it
3. Please provide definitions for all abbreviations before first usage in text
4. Is this a novel approach or similar experiments have been described in literature? Please add relevant references (articles describing this combination of techniques in breast cancer or other cancers and also each component of the technique and their usage in cancer therapy). Comment on results available in literature.
5. An comparison between patients with one treatment and those with multiple treatments would be interesting. Please provide this in terms of tumor shrinkage, return to resectable status and survival. Include details on metastatic disease vs. non metastatic in each group of patients.
6. Adverse reactions should be discussed further and compared to other available treatment options (if possible)
7. Limitation of study and possible future directions for research.
8. In the absence of further study a very definitive conclusion is not possible. The authors should use "seems to be......" in stead of "is"
Comments on the Quality of English LanguageMinor editing required.
Author Response
Reviewer:
The manuscript presented for review is meant to be a "proof-of-concept" article describing the effect of combined reversible electroporation and chemotherapy administered by arterial infusion or isolated perfusion in locally relapsed unresectable breast cancer.
The method presented is intriguing and may provide a safe solution for selected patients in cases where other therapeutic options have been depleted.
My suggestions for improvement are as follows:
1. Minor English proofing needed
2. Please decide on just one abbreviation for circulating tumor cells and use it
3. Please provide definitions for all abbreviations before first usage in text
4. Is this a novel approach or similar experiments have been described in literature? Please add relevant references (articles describing this combination of techniques in breast cancer or other cancers and also each component of the technique and their usage in cancer therapy). Comment on results available in literature.
5. An comparison between patients with one treatment and those with multiple treatments would be interesting. Please provide this in terms of tumor shrinkage, return to resectable status and survival. Include details on metastatic disease vs. non metastatic in each group of patients.
6. Adverse reactions should be discussed further and compared to other available treatment options (if possible)
7. Limitation of study and possible future directions for research.
8. In the absence of further study a very definitive conclusion is not possible. The authors should use "seems to be......" in stead of "is"
Authors answers to the Reviewer:
- We read and corrected the manuscript accordingly
- The term CETC/CTC was chosen to emphasize that the maintrac analysis method specifically detects and quantifies circulating epithelial tumor cells in the blood. In contrast, the abbreviation CTC can also refer to tumor cells from other origins, including hematological tumors but frequently is used in the literature. Therefore we would like to keep this double abbreviation to clarify the source of cells and keep comparability to other literature.
- We added definitions where they were missing
- This is a novel approach. To the best of our knowledge, the combination of arterial infusion or isolated perfusion never has been described before. Due to the lack of knowledge on this combination, we conducted the measurements of circulating tumor cells. This measurements should provide knowledge on the danger of releasing tumor cells from the solid cancer into the blood stream due to electric voltage. Because the chemotherapy is applicated only locoregionally a potential systemic increase of circulating tumor cells after the treatment was to be surveilled.
- Due to the limited number of patients a definitive comparison of subgroups is not appropriate, However we provided a table with all patient characterisitics, number of therapies with electrochemotherapy, tumor sizes before and after completion of electrochemotherapy, reached tumor shrinkage, and overall survival. So far, no correlation of tumor shrinkage or overall survival could be drawn to tumor size or treatment frequencies. This is mainly because of the presence/development of life threatening metastases, which were not in the focus of this study. The aim of this study was mainly the feasibilty and safety of the combination of these methods and the general response behaviour.
- A more detailed description of adverse events has been added to the results section. However a comparison to other methods is difficult as trials with comparably advanced patient groups with equally big tumors are rare.
- Limitations of the study are the small patient number and the heterogeneity of the patients in terms of tumor size and metastatic lesions. The latter sentence was also added to the discussion section.
- We agree to the reviewer in this point. The conclusion text was changed according to the reviewer’s request “is“ has been changed to „seems to be…“
Reviewer 3 Report
Comments and Suggestions for Authors
Please see the attached.

Please check through the whole paper to confirm the gramma is fine. For example, in page 6, “For arterial infusion and isolated abdominal, pelvic, or limb perfusion chemotherapy mostly drugs that also work under hypoxic conditions like cisplatin, doxorubicin and mitomycin are used worldwide.” Missing subject. There are a lot of similar problems in the whole article.
Reviewer 4 Report
Comments and Suggestions for Authors
Comments
Journal: Cancers
Article Type: research
Article ID: cancers-3231008
The paper describes the use of two different methods to improve the treatment of TNBC. Overall, it needs more details and description. The authors must focus on the writing in the TNBC, improve the figures and tables, and provide more details. The current manuscript is not good enough.
Abstract
In general, it is well written, but the language must be moderated, and abbreviations must be used throughout the text.
1. Describe CT
2. If you describe CTC, use it constantly!
3. “huge” and “tremendous” cannot be applied… Change the concepts.
Keywords
1. Do not use huge!
Introduction
The introduction is too poor. Needs a lot more information.
1. If the paper is focused on TNBC, please mention just this breast cancer subtype instead of making general assumptions about breast cancer in general
2. Put more info on the epidemiology of TNBC
3. Put more information on the problems with TNBC
4. Pt more information on the current treatments and problems for TNBC
Materials and Methods
The section has all the information, but needs more details
1. Figure 2 is yours? If not, you have to put the reference.
2. What is the dose of the drugs?
3. A table with patients’ information is needed! Age, TNM, diagnosis, treatments,…
4. EpCAM
Results
The section needs more details. The authors have more results. They must describe them a little bit more
1. Delete the first statement…
2. Figure 4 and 5 are very poor… Improve them
Discussion
Nicely written. However, it needs more references.
Author Response
Reviewer:
Abstract
In general, it is well written, but the language must be moderated, and abbreviations must be used throughout the text.
- Describe CT
- If you describe CTC, use it constantly!
- “huge” and “tremendous” cannot be applied… Change the concepts.
Keywords
- Do not use huge!
Introduction
The introduction is too poor. Needs a lot more information.
- If the paper is focused on TNBC, please mention just this breast cancer subtype instead of making general assumptions about breast cancer in general
- Put more info on the epidemiology of TNBC
- Put more information on the problems with TNBC
- Pt more information on the current treatments and problems for TNBC
Materials and Methods
The section has all the information, but needs more details
- Figure 2 is yours? If not, you have to put the reference.
- What is the dose of the drugs?
- A table with patients’ information is needed! Age, TNM, diagnosis, treatments,…
- EpCAM
Results
The section needs more details. The authors have more results. They must describe them a little bit more
- Delete the first statement…
- Figure 4 and 5 are very poor… Improve them
Discussion
Nicely written. However, it needs more references.
Authors' answers to the Reviewer:
Abstract:
- CT is the abbreviation for computer tomography. the description has been added to the abstract
- The term CETC/CTC was chosen to emphasize that the maintrac analysis method specifically detects and quantifies circulating epithelial tumor cells in the blood. In contrast, the abbreviation CTC can also refer to tumor cells from other origins, including hematological tumors but frequently is used in the literature. Therefore we would like to keep this double abbreviation to clarify the source of cells and keep comparability to other literature.
- We agree to the reviewer and changed this unspecific adjectives to specific descriptions (mostly huge was changed to unresectable)
Kewords:
- see above
Introduction:
- To emphasize the specific characteristics of TNBC we needed some description of the other subtypes as TNBC has no own specific characteristics. The only common characteristic is the lack of the receptors of other subgroups.
- We added more description on TNBC to the introduction, including epidemiology and problems.
- see above
- see above
Materials and Methods:
- Yes, the figure is owned by the authors.
- Drug dosages have been added to the methods section
- a table with patient information has been added to the results section
- EpCAM cell selection and counting was described in more details in the methods section
Results:
- We deleted the first statement as it was a part of the template
- Figure 4 has been changed and contaons more details now. We did not change figure 5 as we think it contains all information that is relevant.
Discussion:
We thank the reviewer for this comment and added more references.
Round 2
Reviewer 2 Report
Comments and Suggestions for Authors
The authors have addressed all my concerns adequately and I believe the manuscript can be published in the current form.
Author Response
Dear Reviewer 2,
we thank you for reviewing our manuscript and for the constructive suggestions you have made. It helped a lot improving the manuscript and we are happy to have fulfilled your suggestions.
the authors
Reviewer 3 Report
Comments and Suggestions for Authors
Thank you for answering all the questions and modified the manuscript as suggested. One more suggestion is that please include the proper description of Figure 1A and 1B. Thank you!
Author Response
Dear Reviewer 3,
we thank you for reviewing our manuscript and for the constructive suggestions you have made. It helped a lot improving the manuscript and we are happy to have fulfilled your suggestions. We have additionally amended a full description to figure 1A and 1B now.
the authors
Reviewer 4 Report
Comments and Suggestions for Authors
The authors replied successfully to all my comments.
Author Response
Dear Reviewer 4,
we thank you for reviewing our manuscript and for the constructive suggestions you have made. It helped a lot improving the manuscript and we are happy to have fulfilled your suggestions.
the authors